# Synaptic Disruption by Soluble Oligomers in Patients with Alzheimer’s and Parkinson’s Disease

**DOI:** 10.3390/biomedicines10071743

**Published:** 2022-07-19

**Authors:** Berenice A. Gutierrez, Agenor Limon

**Affiliations:** Mitchell Center for Neurodegenerative Diseases, Department of Neurology, School of Medicine, University of Texas Medical Branch, Galveston, TX 77555, USA; begutier@utmb.edu

**Keywords:** oligomers, E/I balance, neurodegenerative diseases

## Abstract

Neurodegenerative diseases are the result of progressive dysfunction of the neuronal activity and subsequent neuronal death. Currently, the most prevalent neurodegenerative diseases are by far Alzheimer’s (AD) and Parkinson’s (PD) disease, affecting millions of people worldwide. Although amyloid plaques and neurofibrillary tangles are the neuropathological hallmarks for AD and Lewy bodies (LB) are the hallmark for PD, current evidence strongly suggests that oligomers seeding the neuropathological hallmarks are more toxic and disease-relevant in both pathologies. The presence of small soluble oligomers is the common bond between AD and PD: amyloid β oligomers (AβOs) and Tau oligomers (TauOs) in AD and α-synuclein oligomers (αSynOs) in PD. Such oligomers appear to be particularly increased during the early pathological stages, targeting synapses at vulnerable brain regions leading to synaptic plasticity disruption, synapse loss, inflammation, excitation to inhibition imbalance and cognitive impairment. Absence of TauOs at synapses in individuals with strong AD disease pathology but preserved cognition suggests that mechanisms of resilience may be dependent on the interactions between soluble oligomers and their synaptic targets. In this review, we will discuss the current knowledge about the interactions between soluble oligomers and synaptic dysfunction in patients diagnosed with AD and PD, how it affects excitatory and inhibitory synaptic transmission, and the potential mechanisms of synaptic resilience in humans.

## 1. Introduction

Alzheimer’s disease (AD) and Parkinson’s disease (PD) are among the most prevalent neurodegenerative diseases that shares the misfolding of proteins and synaptic dysfunction as part of their neuropathology. The World Health Organization estimates that 55 million people worldwide live with dementia, of which two-thirds are due to AD [1]. PD is the second most common age-related neurodegenerative disorder after AD with an estimated of up to 10 million people worldwide [2]. Due to the large prevalence of these disorders, the economical and psychological costs on society, caregivers and individuals affected is extremely high. Although large advances in understanding the potential causes of the clinical symptoms at different levels of analysis has been made, there is still the need of effective disease-modifying treatments that can help people with a diagnosis and those at high risk. In this review we will discuss recent studies about the role of oligomers of misfolded proteins on the vulnerability of the distinct types of synapses and the potential mechanisms of synaptic resilience in humans. 

AD is a slowly progressive neurological disease, clinically characterized by a decline in memory, language, and other cognitive skills [3,4]. The diagnosis as recommended by the National Institute on Aging-Alzheimer’s disease Association (NIA-AA) research framework requires the presence of extracellular amyloid plaques and intracellular neurofibrillary tangles under a microscopic examination of several brain regions [5]; thus, the definitive diagnosis of AD is made after death. Alive people with amnestic dementia are usually diagnosed with AD clinical syndrome [6].

The pathological hallmarks of AD are the final expression of a molecular chain of events, including misfolding of proteins that form large polymers of aggregates. Amyloid-beta (Aβ) deposits are the consequence of abnormal cleavage of the Aβ precursor protein (APP), a cell surface receptor thought to regulate neuronal migration during development [7]. APP cleavage can generate amyloidogenic (pathogenic) and non-amyloidogenic (non-pathogenic) Aβ forms. The β-site APP cleaving enzyme I (BACE1) cleaves APP close to its N-terminus between residues M671 and D672 and is responsible for generating the amyloidogenic Aβ form [8]. Abnormal aggregation of Aβ monomers (Aβ40 and Aβ42 with 40 and 42 amino acids) leads to the formation of Aβ oligomers (AβOs) and the establishment of extracellular amyloid plaques that promote neuronal synaptic and cell loss [7]. Additionally, but not necessarily in parallel, tau proteins (ranging from 352 to 421 amino acids in AD), which are part of the cytoskeleton and critical for intracellular transport mechanisms [9,10,11], become hyperphosphorylated causing their detachment from the microtubules and leading to cytoskeletal destabilization. Tau monomers are normally found unfolded and do not form filaments. However, abnormal aggregation into straight and paired helical filaments may occur; this process is driven by a shift from random coil to a β-sheet structure of regions within the second and third repeat domains [12]. When tau monomers detach from the microtubules, they form oligomers (TauO), which are the foundation of the intracellular neurofibrillary tangles that disrupt the synaptic transmission and lead to neuronal death [7]. Interestingly, there is no clear association between the number of Aβ and tau deposits with the severity or duration of the disease; moreover, there are adults with normal cognition and remarkable amounts of those proteins, particularly Aβ [13,14,15].

The ‘core’ of the neuronal lesions in PD is the progressive degeneration of dopamine neurons in the central nervous system (CNS), which accounts for most of the prominent symptoms (slowness of movement, rest tremor, and rigidity) [16,17,18]. The neurodegeneration of PD is driven by accumulation of α-synuclein (αSyn) and the presence of Lewy bodies and dystrophic neurites, in the central and peripheral nervous systems [19,20]. α-syn is an unfolded protein of 140 amino acids that lacks a distinct secondary or tertiary structure, and whose conformational changes generate the insoluble neurotoxic species of PD [8]. The first association between Aβ and AD was suggested by Glenner and Wong [21], while the one between αSyn and PD was done by Goldberg and Lansbury [22]. There is strong evidence of co-occurring pathologies across these neurodegenerative diseases. For example, αSyn accumulation is frequently observed in AD, whereas amyloidosis and tauopathy are also observed in PD, indicating protein-protein interactions and cross-seeding between Aβ, tau and αSyn proteins which promote aggregation and accelerate cognitive impairment [23]. Importantly, the generation and release of all these oligomers to the extracellular space is highly interdependent on the electrical activity of neurons [24,25] and has strong effect on synaptic plasticity.

## 2. Neurodegeneration Driven by Small Soluble Oligomers 

Oligomers are small, soluble protein aggregates which possess unique structural and functional properties. They are intermediary between soluble monomeric proteins and insoluble mature fibrils [26]. For several years, therapeutic research in AD and PD was centered in targeting insoluble fibrillar aggregates of Aβ, tau and αSyn, but recent studies have shown that soluble oligomers are the most toxic species that induce neuronal damage and dysfunction in neurodegenerative disorders [27,28,29,30,31,32,33,34,35,36], suggesting that anti-oligomeric therapeutic strategies would be a better approach to antagonize cognitive deficit symptoms. 

Neurodegeneration driven by AβOs has been experimentally proven using human postmortem brain tissue from subjects clinically diagnosed with AD. AβOs extracted from these subjects have shown to alter long-term potentiation, enhance long term depression, and reduce the dendritic spine density of pyramidal neurons in the hippocampus of control mice [37]. The reduction of spine density is consequence of the loss of spine cytoskeletal proteins, a phenomenon that implicates impairment of memory-related receptors such as NMDA receptors [37,38] (See below for effects of oligomers on synaptic receptors). AβOs also contribute to loss of synaptic markers such as synaptic vesicle-associated membrane protein 2 and post-synaptic density protein 95 [39] indicating reduction of synapses and loss of communication between neurons. The synaptic deterioration manifests with memory and learning impairment as observed in control rats after injection of AβOs from human AD brains [37]. Further, levels of AβOs in fractionated brain homogenates from patients with AD correlate with the severity of cognitive impairment (assessed by Blessed Information-Memory-Concentration and the Mini-Mental State Examination scores) [39]. Nevertheless, AβOs are found in subjects without cognitive impairment; they increase physiologically in old age [40]. The fact that both demented and non-demented patients have increased levels of oligomers does not necessarily mean they have the same oligomeric structure. AβOs organize into dimers, trimers, tetramers, and higher order structures. Three oligomeric structures amyloid-β trimers, Aβ*56 and amyloid-β dimers—have been identified in AD mouse models and humans [40,41]. In a cross-sectional analysis of the Religious Orders Study, two important observations were done: first, although all three AβOs increase in old age, they are always significantly higher in demented compared to cognitively intact patients; and second, the oligomeric form Aβ*56, but not Aβ dimers nor Aβ trimers, correlate positively with soluble pathological tau proteins and negatively with postsynaptic proteins in intact subjects [40]. These results propose different planes of participation of the three oligomeric structures in physiological and pathological processes [40]. For instance, Aβ*56 (56-kDa oligomer) appears to be a major oligomer forming complexes with NMDA receptors and mediating Ca^2+^ influx and activation of Ca^2+^-dependent calmodulin kinase IIα, which in turn induces disarrangement and phosphorylation of tau [40,42]. Interestingly, control subjects or Non-Demented with AD Neuropathology (NDAN)—subjects with histopathological, but not clinical AD—lack binding of AβOs to NMDA receptors or other synaptic structures [37,43], strongly suggesting that binding of AβOs to synapses is an important trigger of synaptic degeneration. 

The relation between oligomer formation and disease state remains controversial. Most studies support the pathogenic role of oligomers in neurodegeneration as mentioned above. For example, elevated levels of plasma AβOs have shown a strong correlation with the cognitive performance in patients with AD (assessed by Mini-Mental State Examination, Cognitive Abilities Screening Instrument, Alzheimer’s Disease Assessment Scale–cognitive portion, and common objects memory test) [44]. Other studies have shown correlation with severity [45,46]. Additionally, in transgenic mice (PS1V97L), inhibition of AβOs showed to improve the memory function [47]. However, a few other groups have found contradictory results. AβOs were not able to induce memory deficit in mutant mice with loss of APP, suggesting that either APP is a key component of cognitive decline or that Aβ aggregates affect cognition by a yet unknown mechanism [48]. Another study showed cognitive improvement in mice with amyloid pathology after lowering the APP/Aβ production, while the amount of AβOs remained unchanged [49]. Altogether, evidence points out that oligomers play a role in the pathophysiology of neurodegenerative disorders; however, at what time of the AD continuum they become clinically significant, and whether they are the unique and most important target still needs to be determined. 

TauOs are present in neurons and astrocytes at early stages of neurodegeneration, they are increased even before the formation of neurofibrillary tangles and clinical manifestations of AD [50,51,52]. As with AβOs, human brain derived TauOs injected in mice impair synaptic plasticity in the hippocampus, and clinically manifest with anterograde memory storage dysfunction [53]. TauOs are not only increased in senile and AD brains, but also, they have been detected in high levels in animal models and brains of individuals with PD, suggesting that TauOs are neurotoxic mediators in synucleinopathies [54,55]. TauOs have been associated with stress granules and molecular markers such as eIF3η, TIA1, PABP and HNRNPA2B1, inducing a translational stress response [56,57]. Moreover, TauOs form a complex with m^6^A and HNRNPA2B1, which is increased up to 5-fold in subjects with AD, meaning that this complex contributes to neurodegeneration [57]. Transactive response-DNA binding protein 43 (TDP-43), has been found in stress granules of patients with frontotemporal dementia and amyotrophic lateral sclerosis, and now it was shown that TDP-43 forms oligomeric assemblies that associate with AβOs and TauOs, suggesting co-partnerships in the pathogeneses of neurodegenerative diseases [58]. Further, TauOs interact with the Musashi family of RNA-binding proteins in AD brains to form nuclear aggregates and induce reduction of LaminB1, leading to nuclear instability and thus, neuronal dysfunction [59]. 

Neurodegeneration driven by αSynOs depends on their interaction with cell membranes. Similar to AβOs and TauOs, αSynOs can perforate the membrane and hence alter the membrane conductance; therefore, formation of ion-permeating pores seems to be a general mechanism to destabilize the cell membrane shared by some oligomeric forms of misfolded proteins. Moreover, perfusion of αSynOs onto hippocampal neurons induce an increase of intracellular calcium level, which supports the idea of strong membrane interactions [60,61]. The elevation of calcium levels is in accordance with the calcium homeostasis dysregulation observed in PD subjects [62,63,64,65]. Importantly, αSynOs contribute significantly to dopaminergic loss and neuronal cell death, which is not observed to be caused by αSyn fibrils, indicating once again that oligomers, but not insoluble fibrils, are the most toxic species in PD and other neurodegenerative disorders [61,66]. 

## 3. Synaptic Dysfunction Leading to Cognitive Impairment

### 3.1. Synapse Loss

It has been well established that synaptic dysfunction occurs in AD and PD. Less synapses are found in postmortem brains of patients with AD and PD in brain regions underlying clinical manifestations of both diseases [67,68]. However, only recently it became possible to evaluate synaptic alterations in alive people by using synaptic positron emission tomography (PET) imaging. Specifically, the PET tracer [^11^C]UCB-J for the synaptic vesicle glycoprotein (SV2A), expressed in all synapses and located in synaptic vesicles at presynaptic terminals, was recently used to detect synaptic alterations in vivo of patients with early AD and PD. In AD, PET imaging of SV2A showed prominent reduction synapses in the hippocampus, followed by the entorhinal cortex, parahippocampal cortex, amygdala, lateral temporal cortex, PCC/precuneus, and lateral parietal cortex, but not in the prefrontal cortex, lateral occipital cortex, medial occipital cortex, or pericentral cortex [69]. The synaptic density reductions were maintained after partial volume correction of the PET images, meaning that the effect is not entirely attributed to loss of gray matter tissue. Importantly, there was a correlation between the reduction of SV2A and cognitive impairment. PET studies correlate with accumulated literature that has consistently shown evidence of synaptic loss across brain regions in AD and other neurodegenerative disorders [70]. In PD, PET imaging showed lower SV2A in the substantia nigra, followed by red nucleus and locus coeruleus as well as other clinically relevant areas [71]. Interestingly, neurocognitive assessment in PD did not correlate with SV2A levels likely because cognitive impairment is milder than that observed in AD and thus the range for correlation is narrower. 

### 3.2. Inflammatory Response Effects on Synapses

Although AD and PD were not originally considered inflammatory disorders, neuroinflammation is a critical component in the pathogenesis and progression of cognitive impairment. Neuroinflammation involves activation of microglia and astrocytes, and the subsequent release of cytokine radicals which lead to synaptic loss and damage [72,73]. Particularly, microglia are pivotal in the control of synapse activity by establishing direct contact with neurons, meaning that an inflammatory process at this level has a negative impact on synaptic surveillance and thus, cognitive function. However, whether neuroinflammation is caused by soluble oligomers, the most toxic components in the pathology of AD and PD, is not clear yet. Here we present current studies addressing the association between soluble oligomers and synaptic dysfunction due to neuroinflammation.

Distinct Aβ conformations seem to trigger different magnitudes of microglial activation. As mentioned before, oligomeric (rather than fibrillary) forms of Aβ, are the most neurotoxic aggregates in AD [37,38,74,75,76]. Thus, it has been investigated in vitro and in vivo whether AβOs are also stronger promoters of glial activation. An in vitro glial cell culture exposed to AβOs and fibrillar-Aβ, demonstrated not only that the pro-inflammatory response of the oligomeric form of Aβ was stronger than its fibrillary counterpart, but also that the response was an M1-like phenotype [77]. Complementarily, a murine study, where brain inflammation was induced by different Aβ42 conformers, showed that the lightest AβOs can activate microglial cells and promote a violent inflammatory response, whereas heavier oligomeric and fibrillary Aβ conformations induced less glial activation and poorer inflammatory responses [78]. Another in vivo model demonstrated AβOs promoted stronger neurotoxicity and inflammatory response mediated by NF-κB, when compared to fibrillar-Aβ [79]. All these studies reinforce the idea that AβOs are the most potent activators of microglial cells, and following studies display how this inflammatory response leads to synaptic disruption and sequential neuronal dysfunction. The inflammatory response followed by synaptic disruption and neuronal loss can be clinically translated as memory, language, and visual perception decline, among other forms of cognitive impairment [80]. In animal models AβOs induce inflammatory signaling leading to this cognitive decline manifestations [81,82,83]. For instance, in an acute experimental model in C57BL/6 mice, memory impairment and inflammation were observed after an intracerebroventricular injection of AβOs, suggesting that oligomers interfere with synaptic transmission necessary to establish new memories; again, the fibrillar-Aβ did not produce this effect [32]. The molecular link between cognitive deficit and neuroinflammation lies in the release of cytokines by microglial cells. In one study, purified AβOs from human AD brain tissue were injected in wild type mice to induce microglial inflammation. The inflammatory response of this model was demonstrated when several cytokines at mRNA and protein levels were identified, including *Ccl3*, *CCl4*, and *Tnf* [84]. Other mechanism underlying AβO-induced microglial activation is explained by TLR-4, which likely induces aberrant TNF-α signaling [85,86]. In support of this deleterious role of the inflammatory response, the cognitive decline, induced by the intracerebral injection of AβOs, was reversed by the administration of anti-inflammatory drugs, doxycycline, and TLR-4 antagonists [87]. In another study of intracerebroventricular injection of AβOs in wild type mice, the complement factors C1q, which initiates the classic complement pathway, and C3, were found elevated at the synapse level, which would explain the synapse loss through microglial activation [88]. In addition to inflammation induced by microglia, astrocytosis is another early phenomenon in AD development, but whether AβOs induce astrocytosis remains to be determined. 

TauOs are the most neurotoxic tau species involved in the development of cognitive impairment [89,90]. They induce neuroinflammation in AD and frontotemporal lobar dementia through interaction with astrocytes and microglia [91]. A model for the toxic relationship between TauOs and inflammation has been proposed, where TauOs through astrocytes and microglia trigger the release of cytokines, RAGE receptors and their ligand HMGB1. Activation of RAGE signals NF-κB and p38-MAPK pathways, which in turn promote hyperphosphorylation of Tau and subsequent aggregation of more oligomers, and thus, neuronal damage or death and a vicious cycle of chronic neuroinflammation [51,91]. Levels of IL-1β and TNF-α increase significantly when microglia cells are exposed to TauOs [92]. 

Inflammatory reaction induced by αSyn can take place in the microglia and astrocytes or have a direct effect on neurons. Upon activation of microglia by αSyn, the microglia release neurotoxic factors, including pro-inflammatory cytokines which may lead to neuronal dysfunction [93,94]. As in Aβ and Tau, αSyn self-aggregates to generate αSynOs, protofibrils and fibrils. It is also well established that αSynOs are the main responsible form for cognitive decline in Lewy body dementia and PD. Whether αSynOs contribute more than fibrillar α-syn to neuroinflammation in PD, in the same way as AβOs and TauOs in AD has been a recent area of interest. One study showed, injection of αSynOs in the brain ventricles of wild type mice caused memory impairment through a TLR-2 dependent mechanism, which is closely associated with activation of glial cells in the hippocampus; and contrary to AβOs, TLR-4 was not involved in memory impairment [95,96]. On an age-depending study in mice, αSynOs induced an inflammatory response in microglia cells through release of TNF-α in adult, but not young mice, mimicking the inflammatory response in PD pathology [97]. Besides TLR-2 [95,98], αSynOs has also been implicated in the TLR-1 pathway, in either case, the pro-inflammatory microglial phenotype leads to translocation of NF-κB and increased production of TNF-α and IL-1β through MyD88 dependent mechanisms [99]. The monomeric and fibrillary forms of αSyns are not able to produce an inflammatory response of microglial cells [95,96,98] nor impair memory of tested mice [95]. The cognitive deficit induced by αSynOs is antagonized with pretreatment of anti-inflammatory drugs [95]. In astrocytes, αSynOs induce neuronal cell death in a TLR-4 dependent mechanism by triggering the production of TNF-α and other cytokines by astrocytes [96]. Neurons alone are also directly susceptible to αSynOs induced TLR4-independent toxicity [96]. Additionally, αSynOs can induce the production of reactive oxygen species (ROS) on the presence of free metal ions, resulting in neuronal death [100]. At the peripheral level, neuroinflammation induced by αSynOs also enhances and aggravates cognitive deficits in mice [101].

Gamma-aminobutyric acid (GABA) plays an important role in the communication between neurons and microglia. Microglia and other CNS cells through altered GABA receptors may lead to impaired signaling and thus, lose communication with neurons [102]. GABAergic signaling in microglia decreases the activity of inflammatory mediators NF-κB and p38 MAP kinase and suppress the release of TNF-α and IL-6 outside synapses [103]. This suggests that GABA receptors may be a potential target to reduce the deleterious effects of oligomers on synaptic dysfunction. In support, enhancement of GABA_A_ receptors through drugs such as carbamazepine, phenytoin, and valproic acid have been found to stabilize intracellular Ca^2+^ levels and thus ameliorate the neurotoxic effects of AβOs [104]. GABAergic signaling across neurons and glial cells that modulate homeostatic plasticity is an area that has been understudied and requires more multidisciplinary efforts from the research community.

### 3.3. Receptors Involved in Synaptic Dysfunction

Accumulated evidence indicates that AβOs directly activates AMPA receptors [105]. AMPA receptors are complex proteins made by the combination of four principal subunits (GluA1-GluA4) [105,106], and co-assembled auxiliary proteins [107,108], that modulate the gating, permeability, and pharmacology of the channel [108,109,110,111,112]. GluA2-lacking AMPA receptors are permeable to Ca^2+^ and its excessive activation leads to Ca^2+^ overload, excitotoxicity, and neurodegeneration [113,114,115,116,117]. Recent evidence from Reinders et al., demonstrated that AβOs cause synaptic failure only in neurons expressing GluA3 subunits [118], and mice with severe AD neuropathology but deficient in GluA3 were cognitively resilient [118], strongly indicating that synaptic vulnerability to AβOs may depend on the stoichiometry of synaptic receptors. This is consistent with human postmortem studies where lower gene expression levels for GluA3 correlated with better cognitive performance in prodromal AD [119]. Similarly, it is increasingly acknowledged that AβOs directly activate heterologously-expressed receptors composed by GRIN1/GluN2A and GRIN1/GluN2B subunits [105,120], which are the most abundant NMDA receptors in mammals’ cortical synapses; however, only the activation of receptors containing GluN2B subunits (GluN2B-NMDA receptors) leads to acute activity-dependent postsynaptic failure [121], Ca^2+^ dysregulation [122], synaptic depression [123,124], and neurotoxicity in in vitro systems [125,126]. Most likely due to the high Ca^2+^ permeability of GluN2B-NMDA receptors [127] and their downstream signaling [128]. The clinical significance of GluN2B is reinforced by a multisite postmortem study showing that lower cortical gene expression of GluN2B correlates with better cognitive performance in people diagnosed with prodromal AD [119]. These parallel lines of evidence strongly suggest that levels of expression of GluN2B-NMDA receptors are correlated with synaptic and neuronal vulnerability. Although little is known about the mechanism by which αSyn produce synaptic dysfunction, Trudler at al., recently showed that αSynOs induce Ca^2+^-dependent release of glutamate from astrocytes leading to a chronic increase of glutamate that activates extrasynaptic NMDA receptors and inducing synaptic loss [129]. αSynOs also bind to NMDA receptors, increasing the synaptic transmission and resulting in membrane damage and LTP impairment [130]. Also, by targeting GluN2A NMDA receptors, αSynOs can induce visual spatial memory impairment [131].

In contrast to AβOs or αSynOs, TauOs have not been demonstrated to directly interact with synaptic receptors. However, tau participates in Aβ mediated toxicity by interacting with Fyn kinase via its amino-terminal projection domain facilitating the NMDA receptors-mediated synaptotoxicity [132]. The role of tau in Aβ toxicity via Fyn-kinase modulation is further supported by studies reporting that absence of tau in dendritic spines prevented the toxic effects of AβOs mediated by GluN2B-NMDA receptors [133], and whereas a reduction in tau levels prevented the cognitive impairment in AD transgenic mice overexpressing Aβ, overexpression of Fyn can enhance their cognitive impairment [134]. Taken together, these results strongly suggest that AβOs and αSynOs may initiate their toxic effects by binding to specific subtypes of AMPA and NMDA receptors and/or the proteins they form complexes with; and tau is an important element for downstream signaling of the neurotoxicity. It follows that, differential expression of those particular targets may provide synaptic protection and underlie cognitive resilience in humans.

### 3.4. Impaired Excitatory/Inhibitory Ratio

Hyperexcitability of cortical and hippocampal circuits and 87-fold increase in seizures incidence in the AD population is well documented [135], particularly in early-onset familial AD [136]. Convulsive seizures occur in approximately 7–21% of sporadic AD patients [137,138], 31% of patients with PS2 mutations [139] and 56% of patients with APP duplications [140]. These data do not account for hidden hyperexcitability status that occurs early in AD pathogenesis [141]. Since oligomers act mostly on excitatory synapses leading to dysfunction first and synaptic loss later, a large question in the field is how, overall reduction of excitatory inputs leads to hyperexcitability in the AD brain. Although the causes of network hyperactivity are still under investigation by many labs; early studies in animal models suggest that impaired inhibition is a potential mechanism for network hyperactivity [142,143]. Impairment of interneuron activity with changes in their intrinsic properties have been reported in the mice models of amyloidosis [144]. Interneuron deficits reduces neurogenesis and neuronal maturation in the hilus of the hippocampus [145] and leads to age and tau-dependent learning and memory deficits [146]. Potentiation of GABA receptors by pentobarbital restores some of the deficits observed by GABAergic impairment [145]. Our initial studies transplanting human receptors and recording their electrical activity observed a dramatic reduction of GABA_A_ receptors in AD [147]. This severe reduction of gene expression was later confirmed by other groups using high throughput microarray technology [148] and provided evidence that in addition to excitatory synaptic loss, inhibitory synapses were also affected in AD. 

Because the abundance, activity, and strength of excitatory and inhibitory synapses in the neocortex are highly correlated in their amplitude and time domains [149,150,151,152]. The “global E/I ratio” defined as equal average amounts of postsynaptic AMPA and GABA_A_ receptor activities is essential for maintaining stability of cortical neurons [150,153] and is tightly regulated within a narrow range by inhibitory plasticity following any excitatory change produced by ongoing sensory experiences and activity-dependent plasticity [154]. Taking in consideration the synaptic alterations found in AD, the immediate question is how the global E/I ratio is affected in AD? Interestingly, functional alterations in the default mode network (DMN) correlate with cognitive impairment in AD and it is early affected in AD [149]. The DMN spans several brain regions including the parietal cortex, importantly, baseline DMN activity is increased in AD and fails to deactivate during cognitive tasks, suggesting that the DMN is abnormally and continuously hyperactive in AD [149,155]. By using fluorescence deconvolution tomography (FDT), where automated systems are used to quantify 30,000 immunolabeled elements within the size constraints of synapses from 3D reconstructions of image z-stacks [156,157,158,159], it was found a larger E/I ratio (il-PSD-95 for excitatory/il-gephyrin for inhibitory synapses), in AD compared to non-demented controls. Therefore, our results indicate an overlooked link between increased regional synaptic E/I ratio, cortical hyperexcitability, and a potential enhancement of activity-dependent amyloidosis [160,161]. The pro-excitatory E/I ratio in AD was confirmed by directly recording the activity of AMPA and GABA_A_ receptors microtransplanted from AD parietal cortex [162]. Importantly the E/I ratio was preserved in NDAN people indicating that preservation of the E/I correlates with the preservation of cognition even in presence of neuropathologic change [163].

Taken together, these data suggest that hyperexcitability in AD patients result from impaired GABAergic inhibition which leads to cortical excess of excitatory synaptic inputs. Notably, patients expressing the homomeric APOE4 allele are at the highest genetic risk to develop late-onset AD, and GABAergic interneurons and synapses are also deficient in patients with APOE4 AD. Bumetanide is a diuretic acting on chloride transporters modulating the E/I ratio via pathways enriched in GABAergic signaling. Importantly, bumetanide was found to be protective against AD as evaluated in mouse models, and more importantly through an electronic health record, which showed that the prevalence of AD in patients older than 65 years old was significantly reduced in those taking bumetanide [164]. 

In PD, there is also evidence of E/I imbalance, however the effects are more brain region specific. Loss of substantia nigra pars compacta (SNpc) dopaminergic neurons lead to hyperactivity of the globus pallidum and excessive inhibition onto the motor thalamus, this ultimately leads to the abnormal movement manifestation of PD [165]. While it is still not clear how aSynOs relate to the loss of SNpc, it is clear that a complex series of pathways interact to produce early dysfunction of SNpc [166]. Recent studies in human organoids with mutations linked to PD show that the E/I is altered in PD with lower inhibition and reduced levels of neurosteroid allopregrananolone even before the presence of neuropathology is observed [167]. This indicates that E/I alterations precede pathology, and its correction may have disease-modifying effects. 

## 4. Conclusions

Oligomeric forms of Aβ, tau and aSyn are the most toxic species affecting synapses leading to synaptic dysfunction and altered neuronal communication in brain regions vulnerable to the neuropathology. The effects of oligomers precede the presence of deposits and seem to be associated to early changes in excitatory and inhibitory synapses. Therefore, oligomers seem to produce a “double hit” on synapses (Figure 1). First, they lead to calcium dys-homeostasis by binding directly to excitatory receptors and leading to a first wave of hyperexcitability, then producing GABAergic dysfunction by a mechanism that is still not understood, which leads to a second chronic wave of hyperexcitablity that ultimately leads to neuronal loss and hypoactivity. Understanding the regional and temporal relationships between oligomers, synaptic targets and E/I balance is a critical need in the field.

## Figures and Tables

**Figure 1 biomedicines-10-01743-f001:**
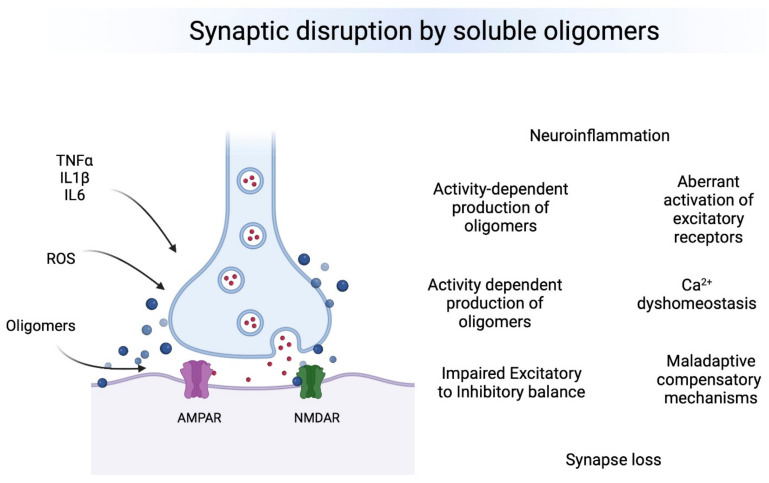
Overview of major effects of toxic oligomers in synapses. Left. Neuroinflammatory and Reactive Oxygen Species (ROS) participate in the production and the effects of toxic oligomers on synapses. Right, Major synaptic effects on synapses. It is still not clear what is the chronological order of events, but each one influence the others and some of them are happening simultaneously at brain regions vulnerable to AD pathology.

## Data Availability

Not applicable.

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
