# Peer review of "Synaptic Disruption by Soluble Oligomers in Patients with Alzheimer’s and Parkinson’s Disease"

_biomedicines, 2022, doi:10.3390/biomedicines10071743_

Round 1

Reviewer 1 Report

In the submitted review, the authors have done an updated and good review on the effect of soluble oligomers in synaptic dysfunctions. Overall, this review is well-written and shed some light on this field. I only have a few suggestions.

-Section 2, line 105: please, define β*56

-Section 3.1 (SV2A PET imaging results): how can be sure that the reduction in SV2A signal is due to a synaptic loss and not an indirect effect  from the cell death that occurs in these disorders?

-Section 3.2, line 245-250. It is not clear the relation of GABAergic signaling and inflammation. Could you clarify it?

-Section 34, line 356. What is the reference for this study?

Finally, It would be convenient to include a scheme that summarizes the effects of the Aβ, tau and syn oligomers according to the different affected molecules and processes (neurodegeneration,synaptic loss, inflammatory response, synaptic dysfuntion...)

Author Response

We greatly thank the reviewer for the helpful comments. Please find below our responses to the comments below:

-Section 2, line 105: please, define β*56

We have defined the term in this new manuscript

-Section 3.1 (SV2A PET imaging results): how can be sure that the reduction in SV2A signal is due to a synaptic loss and not an indirect effect  from the cell death that occurs in these disorders?

This is an excellent point, we have added the following paragraph to section 3.1:

“The synaptic density reductions were maintained after partial volume correction of the PET images, meaning that the effect is not entirely attributed to loss of gray matter tissue. Importantly, there was a correlation between the reduction of SV2A and cognitive impairment. PET studies correlate with accumulated literature that has consistently shown evidence of synaptic loss across brain regions in AD and other neurodegenerative disorders [71]

-Section 3.2, line 245-250. It is not clear the relation of GABAergic signaling and inflammation. Could you clarify it?

We apologized for the lack of clarity. We have added the following paragraph:

 “Gamma-aminobutyric acid (GABA) plays an important role in the communication between neurons and microglia, which are immune cells of the CNS capable of orchestrating a potent inflammatory response. Microglia and other CNS cells through altered GABA receptors may lead to impaired signaling and thus, lose communication with neurons [103]. GABAergic signaling in microglia decreases the activity of inflammatory mediators NF-κB and p38 MAP kinase and suppress the release of TNF-α and IL-6 outside synapses [104]. This suggests that GABA receptors may be a potential target to reduce the dele-terious effects of oligomers on synaptic dysfunction. In support, enhancement of GABAA receptors through drugs such as carbamazepine, phenytoin, and valproic acid have been found to stabilize intracellular Ca2+ levels and thus ameliorate the neurotoxic effects of AβOs [105].”

-Section 34, line 356. What is the reference for this study?

We have added the missing reference

Finally, It would be convenient to include a scheme that summarizes the effects of the Aβ, tau and syn oligomers according to the different affected molecules and processes (neurodegeneration,synaptic loss, inflammatory response, synaptic dysfuntion...)

Thank you for the comment, we include now a figure summarizing the effects of oligomers

Reviewer 2 Report

The manuscript presented by authors Berenice A. Gutierrez and Agenor Limon is a review of the literature regarding the effects of oligomeric forms of proteins involved in neurodegenerative diseases. the authors' focus is directed toward better summarizing the effects of these oligomers at the neuronal level. 

The manuscript is well written and detailed. I just have minor comments and suggestions for the authors.

- I suggest authors include in the introduction section more information regarding Alzheimer's disease as a misfolding disease beyond amyloid-beta and tau.

-How oligomeric formation can be seen in the clinical continuum of Alzheimer's disease. There is a correspondence between the clinical phenotypes and the start spreading of oligomers? I suggest authors to better discuss this point in the manuscript. 

Overall I found the work well written and detailed, of broad interest to a wide audience, thus I suggest the manuscript be published. 

Author Response

The manuscript presented by authors Berenice A. Gutierrez and Agenor Limon is a review of the literature regarding the effects of oligomeric forms of proteins involved in neurodegenerative diseases. the authors' focus is directed toward better summarizing the effects of these oligomers at the neuronal level. 

The manuscript is well written and detailed. I just have minor comments and suggestions for the authors.

We greatly thank the reviewer for the helpful comments. Please find below our responses to the comments below:

- I suggest authors include in the introduction section more information regarding Alzheimer's disease as a misfolding disease beyond amyloid-beta and tau.

Thank you for the suggestion, in response to this we have expanded the information in the introduction and also across multiple paragraphs (marked in red).  We believe the review is more complete in this revised manuscript

-How oligomeric formation can be seen in the clinical continuum of Alzheimer's disease. There is a correspondence between the clinical phenotypes and the start spreading of oligomers? I suggest authors to better discuss this point in the manuscript. 

We thank the reviewer for the suggestion, we have added the following paragraph:

“The relation between oligomer formation and disease state remains controversial. Most studies support the pathogenic role of oligomers in neurodegeneration as mentioned above. For example, elevated levels of plasma AβOs have shown a strong correlation with the cognitive performance in patients with AD (assessed by Mini-Mental State Examination, Cognitive Abilities Screening Instrument, Alzheimer’s Disease Assessment Scale–cognitive portion, and common objects memory test) [45]. Other studies have shown correlation with severity [46,47]. And in transgenic mice (PS1V97L), inhibition of AβOs showed to improve the memory function [48]. However, a few other groups have found contradictory results. AbOs were not able to induce memory deficit in mutant mice with loss of APP, suggesting that either APP is a key component of cognitive decline or that Aβ aggregates affect cognition by a yet unknown mechanism [49]. Another study showed cognitive improvement in mice with amyloid pathology after lowering the APP/Aβ production, while the amount of AβOs remained unchanged [50]. Altogether, evidence points out that oligomers play a role in the pathophysiology of neurodegenerative disorders, however at what time of the AD continuum they become clinically significant, and whether they are the unique and most important target still needs to be determined”

Overall I found the work well written and detailed, of broad interest to a wide audience, thus I suggest the manuscript be published.